# LDL-C:HDL-C ratio and common carotid plaque in Xinjiang Uygur obese adults: a cross-sectional study

Qiang Zhao,[1,2] Fen Liu,[1,2] Ying-Hong Wang,[1,2,3] Hong-Mei Lai,[4] Qian Zhao,[1,2] Jun-Yi Luo,[1,2] Yi-Tong Ma,[1,2] Xiao-Mei Li,[1,2] Yi-Ning Yang[1,2]

QZ and FL contributed equally.

[1]Department of Cardiology, First Affiliated Hospital of Xinjiang Medical University, Urumqi, China
[2]Xinjiang Key Laboratory of Cardiovascular Disease Research, Urumqi, China
[3]Department of Health Management and Physical Examination Center, First Affiliated Hospital of Xinjiang Medical University, Urumqi, China
[4]Department of Cardiology, People's Hospital of Xinjiang Uygur Autonomous Region, Urumqi, China

**Correspondence to**
Professor Yi-Ning Yang;
yangyn5126@163.com

## ABSTRACT

**Objective** The aim of this study was to explore the relationship between low-density lipoprotein cholesterol:high-density lipoprotein cholesterol (LDL-C:HDL-C) ratio and common carotid atherosclerotic plaque (CCAP) among obese adults of Uygur community in Xinjiang, China.

**Design** A hospital-based cross-sectional study.

**Setting** First Affiliated Hospital of Xinjiang Medical University.

**Participants** A total of 1449 obese adults of Uygur population who were free of coronary artery disease were included in our study from 1 January 2014 to 31 December 2016.

**Methodology** Lipid profiles, other routine laboratory parameters and intima-media thickness of the common carotid artery were measured in all participants. Multivariate logistic regression analysis was used to examine the association between LDL-C:HDL-C ratio and CCAP.

**Results** Four hundred and fifteen (28.64%) participants had CCAP. Participants with CCAP had significantly higher LDL-C:HDL-C ratio compared with those without CCAP (3.21 [2.50, 3.88] vs 2.33 [1.95, 2.97], p<0.001). Multivariate logistic regression analysis showed high LDL-C:HDL-C ratio as independent predictor of CCAP after adjusting for conventional cardiovascular risk factors. The top LDL-C:HDL-C ratio quartile (≥3.25) had an OR of 9.355 (95% CI 6.181 to 14.157) compared with the bottom quartile (<2.07) of LDL-C:HDL-C ratio (p<0.001) after adjustment for age, body mass index, smoking, diabetes mellitus and serum level of total cholesterol.

**Conclusion** CCAP is highly prevalent in Uygur obese adults. A high LDL-C:HDL-C ratio is an independent predictor of CCAP. It may help identify obese individuals who are at high risk of CCAP and who may benefit from intensive LDL-lowering therapy.

## INTRODUCTION

Prevalence of obesity has been increasing in China during the past few decades. Presently, overweight or obese people account for one-third of Chinese adults.[1 2] Cardiometabolic risk factors including hypertension, dyslipidaemia and type 2 diabetes mellitus (T2DM) are more prevalent in obese adults,

### Strengths and limitations of this study

► The survey sample was demographically representative of Uygur obese adults aged 18–80 years in Xinjiang.
► This is the first study to date that investigate the association of low-density lipoprotein cholesterol:high-density lipoprotein cholesterol (LDL-C:HDL-C) ratio and common carotid plaque in Uygur obese adults.
► The is a cross-sectional study, we cannot show causality between common carotid plaque and the relevant risk factors.
► The cross-sectional design did not allow us to examine the effects of LDL:HDL-C ratio on the progression of carotid plaque.

partly explaining the association between obesity and increased cardiovascular diseases (CVD).[3–5] Recent data demonstrated that obese individuals who were even metabolically healthy had a higher risk of CVD including coronary artery disease (CAD), heart failure and cerebrovascular disease compared with the normal-weight metabolically healthy individuals.[6] Obesity has been identified as an independent risk factor for CVD and ultimately death,[3 7] and it is emerging as a major public health challenge in China.

Atherosclerosis (AS) is regarded as the primary cause of mortality and morbidity in CVD.[8] It is a well-known fact that AS is a systemic process beginning at early childhood and typically remains asymptomatic until later in life.[9] Approximately 30% of first acute cardiovascular events are fatal, hence the prevention of CVD ought to be directed towards the risk factors including the silent preclinical phase of the process. Using a non-invasive ultrasound biomarker of AS, the carotid intima-media thickness (CIMT) has been demonstrated as a reliable substitute of determining risk of cardiac events in asymptomatic adults.[10 11] Of all the segments of the carotid artery used in

CIMT, the common carotid artery (CCA) is the easiest to image and got the highest reproducibility. Data from the Atherosclerosis Risk in Communities study showed that CCA-CIMT detection was more reliable than measuring CIMT of all carotid artery segments, providing a good marker for CAD risk prediction.[12] Prior researchers have described that the presence of common carotid atherosclerotic plaque (CCAP) was correlated with the increased prevalence of CAD, myocardial infarction, stroke and CVD mortality beyond traditional cardiovascular risk factors.[11–15] Thus, early detection and initiation of treatment of obese patients with CCAP might improve the prevention of cardiovascular events.

Many cohort and nested case–control studies showed that obesity is closely related with high prevalence of dyslipidaemia, characterised by increased plasma triglycerides (TG), total cholesterol (TC), and low-density lipoprotein cholesterol (LDL-C), together with lower high-density lipoprotein cholesterol (HDL-C) levels.[16–19] Previous studies have established the important role of LDL-C in the pathogenesis of AS. LDL-C has been used to monitor the lipid-lowering response to treatments in almost all patients with CVD. However, a number of clinical trials showed 60%–70% of CVD events continue to occur despite LDL-lowering therapy, which may be due to residual risks associated with lipid abnormalities especially atherogenic dyslipidaemia, including increased TG levels and reduced HDL-C levels.[20 21] Therefore, a need for new targets to compliment the measures of LDL-C lowering has been suggested. Recent studies identified LDL-C:HDL-C ratio as a better biomarker to predict both AS progression and CVD than LDL-C or HDL-C alone.[22–25] However, data on the associations between LDL-C:HDL-C ratio and CCAP among obese population are limited. Thus, the present study was designed to explore the relationship between the level of LDL-C:HDL-C ratio and CCAP among Uygur obese adults in Xinjiang to prove this fact.

## SUBJECTS AND METHODS

### Study design and participants

This is a cross-sectional study conducted at the First Affiliated Hospital of Xinjiang Medical University from 1 January 2014 to 31 December 2016 in Xinjiang, China.

The study population was signed-up from the outpatients and inpatients admitted to the cardiology department and participants in routine health examinations at the health physical examination centre of the First Affiliated Hospital of Xinjiang Medical University. Eligible participants were obese Uygur adults aged 18–80 years who were free of clinically evident CAD, which was defined as absence of any previous myocardial infarction, previous or current angina with objective evidence of atherosclerotic CAD or a previous coronary revascularisation procedure. The major exclusion criteria included history of physician-diagnosed CAD and stroke, stenosis of carotid artery >30% on angiography, heart failure, congenital heart disease, current cancer, chronic kidney or liver diseases and thyroid diseases.

### Ethical statement

This study was approved. All participants provided written informed consent.

### Risk factors and definitions

Thorough medical history was obtained and a routine clinical examination including physical examination, blood biochemical examination and carotid ultrasonography was undertaken in all participants. Data on age, gender, cigarette smoking status, body mass index (BMI), systolic blood pressure, diastolic blood pressure, medical history including hypertension, diabetes mellitus (DM), previous medication, creatinine level, uric acid, high-sensitivity C-reactive protein (hs-CRP) and lipid profiles were recorded.

Hypertension was defined as having a history of hypertension and/or an average systolic blood pressure ≥140 mm Hg and/or an average diastolic blood pressure ≥90 mm Hg or if participants were taking antihypertensive medication.[26] The presence of DM was based on a history of diabetes defined as either a current or previous fasting plasma glucose level of 7 mmol/L or higher or when participants were taking antidiabetic medication.[27] BMI was calculated by dividing the weight in kilograms by the height in metres square. Obesity was defined as BMI ≥28 kg/m$^2$ by Chinese standards.[2] Smoking was labelled if participant was currently smoking or had smoked in the previous 6 months.

### Lipid profile analyses

Overnight fasting venous blood samples were collected from all subjects for the assessment of routine laboratory parameters. Laboratory tests were performed at the core laboratory of the First Affiliated Hospital of Xinjiang Medical University. Fasting plasma glucose, blood urea nitrogen, creatinine, uric acid and hs-CRP were determined by the standard methods. TG, TC, LDL-C and HDL-C levels were directly measured using the homogeneous enzymatic colorimetric assay (Roche Diagnostics) on an automatic analyzer (Cobas 8000). Dyslipidaemia was calculated on the basis of having at least one of the following conditions: TC concentration >6.22 mmol/L, LDL-C concentration >4.14 mmol/L, HDL-C concentration <1.04 mmol/L, TG concentration >2.26 mmol/L or a history of use of lipid-lowering drugs.[28]

### Carotid ultrasonography assessments

The Acuson Sequoia 512 instrument ultrasound mainframes with a 7.5-Mhz linear array transducer (Siemens AG, Erlangen, Germany) was used to acquire ultrasonographic images of both the right and left CCAs. Intima-media interface lines were manually traced as continuous lines by certified sonographers. The IMT was measured over such a segment of the CCA that was 1 cm long, located approximately 0.5 cm below the carotid artery bulb, and considered not to contain any plaque. It was measured as the distance between the media-adventitia and intima-lumen interfaces.[27] We performed three views on each side of CCAs to determine the maximal and mean CIMT. The presence of CCAP was defined as a CIMT >1.5 mm,

or a focal structure encroaching into the arterial lumen of at least 0.5 mm, or 50% of the surrounding CIMT value.[29] Carotid ultrasonography was performed and evaluated by two independent ultrasonologists who were unaware of the clinical and laboratory data of any of the participants. A third sinologist was consulted if there was a discrepancy between the diagnostic results of the two ultrasonologists. Consensus was then accomplished in all patients.

## Statistical analysis

Statistical analyses were conducted using SPSS V.17.0 (SPSS, Chicago, Illinois, USA). Normally distributed continuous data were presented as mean±SD and compared using unpaired t-tests, while non-normally distributed data were summarised as median with IQR and compared using Mann-Whitney or Kruskal-Wallis tests. Categorical variables were summarised in numbers and percentages and compared between groups with $\chi^2$ tests. Subjects were also classified according to LDL-C:HDL-C ratio quartiles, and the clinical characteristics were analysed using Mann-Whitney tests or $\chi^2$ tests. For subsequent pairwise comparisons, a two-sided Bonferroni p value of 0.05 divided by 6 (0.0083) was used to establish statistical significance. LDL-C:HDL-C ratio was included in the model as a quartile categorical variable. Univariate logistic regression model was used to estimate the independent predictors of CCAP. After performing univariate analysis, significant variables (age, BMI, smoking, DM, TC, and LDL-C:HDL-C ratio quartiles) were used in multivariate logistic regression analysis. Area under the receiver operating characteristic (ROC) curve (AUC) and 95% CI was calculated. A two-tailed value of p<0.05 was considered significant.

## Patient and public involvement

The study was designed to investigate the association of lipid ratio and common carotid artery plaque. However, no patients or members of the public were included in the design, recruitment or conduct of the study. The results of measurements would be disseminated to participants after the study which was completed by the study team. The burden of the intervention will not be assessed by patients themselves.

## RESULTS

A total of 1449 participants with a mean age of 53.81±11.52 years having no history of CAD were recruited in this study. All participants had grouped according to the presence of CCAP. There were 415 (28.64 %) participants in the CCAP group (mean age 55.29±11.04 years and 53.98% were men) and 1034 participants (71.36%) in the non-CCAP group (mean age 53.22±11.67 years and 51.93% were men).

### Baseline characteristics of the study population

The baseline characteristics of all participants according to CCAP status were shown in table 1. There was no significant difference between CCAP group and non-CCAP group with respect to gender, systolic blood pressure, diastolic blood pressure, prevalence of hypertension and previous medications. The participants in the CCAP group were more likely

| Table 1 Baseline characteristics of study population according to CCAP status | | | | |
|---|---|---|---|---|
| **Characteristics** | **Overall (n=1449)** | **CCAP group (n=415)** | **Non-CCAP group (n=1034)** | **P values** |
| Demographics | | | | |
| Gender | | | | 0.482 |
| Women, n (%) | 688 (47.48) | 191 (46.02) | 497 (48.07) | |
| Men, n (%) | 761 (52.52) | 224 (53.98) | 537 (51.93) | |
| Age, years | 53.81±11.52 | 55.29±11.04 | 53.22±11.67 | 0.002 |
| Vital signs | | | | |
| Body mass index, kg/m$^2$ | 30.12 (28.90, 32.32) | 30.39 (29.06, 32.74) | 30.10 (28.76, 32.01) | 0.027 |
| Systolic blood pressure, mm Hg | 130.19±14.13 | 130.99±12.94 | 129.88±14.58 | 0.176 |
| Diastolic blood pressure, mm Hg | 80.98±8.42 | 80.67±8.10 | 81.11±8.54 | 0.369 |
| Medical histories | | | | |
| Smoking, n (%) | 246 (16.98) | 90 (21.69) | 156 (15.09) | 0.002 |
| Hypertension, n (%) | 444 (30.64) | 123 (29.64) | 321 (31.04) | 0.600 |
| DM, n (%) | 324 (22.36) | 133 (32.05) | 191 (18.47) | <0.001 |
| Previous Medications | | | | |
| Antihypertensive drugs, n (%) | 311 (21.46) | 83 (20.00) | 228 (22.05) | 0.390 |
| Antidiabetic agents, n (%) | 147 (10.14) | 51 (12.29) | 96 (9.28) | 0.087 |
| Statins, n (%) | 146 (10.08) | 37 (8.92) | 109 (10.54) | 0.353 |

CCAP, common carotid atherosclerotic plaque; DM, diabetes mellitus.

**Table 2** Laboratory and carotid ultrasonography parameters of study population according to CCAP status

| Characteristics | Overall (n=1449) | CCAP group (n=415) | Non-CCAP group (n=1034) | P values |
|---|---|---|---|---|
| Laboratory measurements | | | | |
| White cell count, $10^9$/L | 6.82±1.87 | 6.71±1.92 | 6.87±1.86 | 0.147 |
| Platelet count, $10^9$/L | 212.13±50.79 | 213.45±51.19 | 211.60±50.65 | 0.531 |
| Haemoglobin, g/L | 137.02±15.71 | 136.49±15.42 | 137.24±15.82 | 0.408 |
| Fasting plasma glucose, mmol/L | 5.28 (4.66, 5.93) | 5.73 (5.17, 6.48) | 5.08 (4.59, 5.65) | <0.001 |
| TG, mmol/L | 1.60 (1.11, 2.15) | 1.53 (1.14, 2.00) | 1.62 (1.10, 2.16) | 0.375 |
| TC, mmol/L | 4.89±0.87 | 5.11±0.84 | 4.80±0.87 | <0.001 |
| LDL-C, mmol/L | 3.04±0.66 | 3.44±0.69 | 2.88±0.58 | <0.001 |
| HDL-C, mmol/L | 1.16 (1.00, 1.35) | 1.10 (0.92, 1.28) | 1.19 (1.03, 1.38) | <0.001 |
| LDL-C:HDL-C ratio | 2.52 (2.07, 3.25) | 3.21 (2.50, 3.88) | 2.33 (1.95, 2.97) | <0.001 |
| Dyslipidaemia, n (%) | 698 (48.17) | 267 (64.34) | 431 (41.68) | <0.001 |
| hs-CRP, mg/L | 2.66 (1.27, 4.42) | 2.71 (1.33, 4.35) | 2.64 (1.25, 4.44) | 0.734 |
| Blood urea nitrogen, mmol/L | 4.87 (4.10, 5.80) | 4.97 (4.10, 5.90) | 4.80 (4.10, 5.71) | 0.284 |
| Creatinine, µmol/L | 70.90±15.34 | 71.82±15.68 | 70.54±15.19 | 0.150 |
| Uric acid, mg/dL | 5.16±1.40 | 5.23±1.43 | 5.13±1.39 | 0.214 |
| Glutamic-pyruvic transaminase, U/L | 27.56±15.64 | 28.44±16.15 | 27.21±15.42 | 0.179 |
| Glutamic-oxaloacetic transaminase, U/L | 19.80 (16.50, 24.21) | 19.60 (16.60, 24.20) | 19.80 (16.50, 24.23) | 0.850 |
| Ultrasonographic parameters | | | | |
| Maximal CCA-IMT, mm | 1.10 (0.80, 1.40) | 1.60 (1.50, 1.80) | 0.90 (0.70, 1.10) | <0.001 |
| Mean CCA-IMT, mm | 0.80 (0.60, 1.10) | 1.30 (1.10, 1.40) | 0.80 (0.60, 0.90) | <0.001 |
| Unilateral plaque, n (%) | 227 (15.67) | 227 (54.70) | – | – |
| Bilateral plaque, n (%) | 188 (12.97) | 188 (45.30) | – | – |
| Maximal CCA-IMT >0.9 mm, n (%) | 429 (29.61) | – | 429 (41.49) | – |

CCA-IMT, common carotid artery intima-media thickness; CCAP, common carotid atherosclerotic plaque; HDL-C, high-density lipoprotein cholesterol; hs-CRP, high-sensitivity C-reactive protein; LDL-C, low-density lipoprotein cholesterol; TC, total cholesterol; TG, triglyceride.

to be older with higher BMI and had higher prevalence of diabetes and smoking than those in the non-CCAP group (all p<0.05).

## Laboratory and carotid ultrasonography parameters of study population

Laboratory and carotid ultrasonography parameters according to CCAP status were summarised in table 2. Out of 1449 participants, 48.17% of them had dyslipidaemia. No significant differences were observed in the serum levels of TG, hs-CRP, blood urea nitrogen, creatinine, uric acid and transaminases between the CCAP and non-CCAP group. Participants with CCAP had significantly higher concentrations of fasting plasma glucose, TC, LDL-C and higher prevalence of dyslipidaemia as well as lower HDL-C level than those in the non-CCAP group. Furthermore, the LDL-C:HDL-C ratio was significantly higher in the CCAP group (3.21 [2.50, 3.88] vs 2.33 [1.95, 2.97], p<0.001) (figure 1).

Out of 415 participants in the CCAP group, 54.70% participants had unilateral and 45.30% participants had bilateral plaques. The maximal and mean CCA-IMT were 1.60 mm (IQR 1.50–1.80) and 1.30 mm (IQR 1.10–1.40) in the CCAP group, respectively. In the non-CCAP group, 41.49% participants had maximum CCA-IMT higher than 0.90 mm, which represented carotid intimal thickening, and the maximal and mean CCA-IMT were 0.90 mm (IQR 0.70–1.10) and 0.80 mm (IQR 0.60–0.90), respectively. Participants in the CCAP group had higher maximal and mean CCA-IMT than those in non-CCAP group (all p<0.001).

As shown in table 3, participants' baseline characteristics were presented by LDL-C:HDL-C ratio quartiles. There were no significant differences in gender, age, prevalence of hypertension and use of drugs between each quartile. Compared with participants in the first quartile of LDL-C:HDL-C ratio, those in the fourth quartile were more frequently smokers, and had higher prevalence of DM and dyslipidaemia. They also had higher prevalence of CCAP and maximal and mean CCA-IMT.

## ASSOCIATION OF LDL-C/:HDL-C RATIO AND CCAP

Univariate and multivariate logistic regression models of the associations between LDL-C:HDL-C ratio and

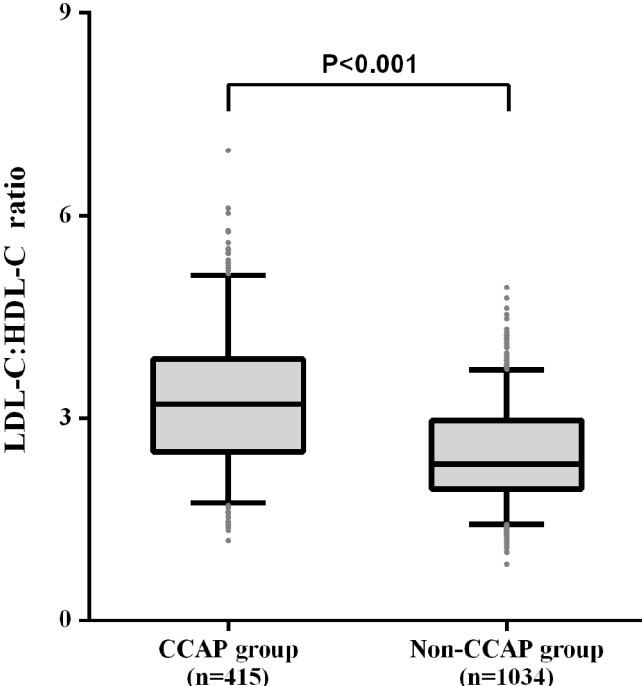

**Figure 1** Comparison of low-density lipoprotein cholesterol:high-density lipoprotein cholesterol (LDL-C:HDL-C) ratio between groups. The level of LDL-C:HDL-C ratio in common carotid atherosclerotic plaque (CCAP) group was significantly higher than non-CCAP group (p<0.001).

CCAP were described in table 4. To determine the linear trends of the risk, we explored the ORs of CCAP by LDL-C:HDL-C ratio quartile groups, with the first quartile serving as the reference category. Entering LDL-C:HDL-C ratio as quartiles to the CCAP univariable model revealed that the top quartile (fourth quartile, ≥3.25) had an OR of 11.464 (95% CI 7.644 to 17.193) compared with the bottom quartile (first quartile, <2.07) (p<0.001). LDL-C (OR 3.012, 95% CI 2.274 to 3.917, p<0.001) and HDL-C (OR 0.222, 95% CI 0.145 to 0.340, p<0.001) were also found to be independent predictors of CCAP in univariate analysis. Other significant correlates of CCAP were age (>45 years, every 10 years), BMI, smoking, DM and TC (all p<0.05). Because LDL-C:HDL-C ratio integrates both LDL-C and HDL-C, and is strongly correlated with the LDL-C and HDL-C, only LDL-C:HDL-C ratio was involved into multivariate analysis.

After adjusting for age, BMI, smoking, presence of DM and mean TC level, the association between LDL-C:HDL-C ratio and CCAP still remained significant (OR (95% CI): second quartile, 2.184 (1.404 to 3.399); third quartile, 3.556 (2.333 to 5.421); top quartile, 9.355 (6.181 to 14.157)). The multivariable-adjusted ORs of CCAP increased continuously and linearly, and statistical significance was observed from the second quartile of the LDL-C:HDL-C ratio. The top LDL-C:HDL-C ratio quartile had an OR of 9.355 (95% CI 6.181 to 14.157) compared with the bottom quartile (p<0.001) after adjustment for age, BMI, smoking, DM and TC.

The ROC curve of LDL-C:HDL-C ratio for predicting CCAP was listed in figure 2. The AUC with LDL-C:HDL-C ratio used to detect CCAP was 0.748 (95% CI 0.720 to 0.776, p<0.001).

## DISCUSSION

In the present study, we examined the associations between LDL-C:HDL-C ratio and CCAP in Uygur obese adults. The main findings of our study are as follows: CCAP is highly prevalent in Uygur obese adults. We observed remarkable differences in the risk of CCAP among subgroups as defined by LDL-C:HDL-C ratio levels, the lowest rate of CCAP (9.67%) was observed in the lowest LDL-C:HDL-C ratio (first quartile) subgroup, whereas the highest rate (55.10%) was observed in the highest LDL-C:HDL-C ratio (fourth quartile) subgroup. We found that LDL-C:HDL-C ratio is an independent predictor of CCAP after adjusting for possible confounding factors including mean TC level. There was a positive association between the increase in LDL-C:HDL-C ratio and the prevalence of CCAP. These findings, if confirmed, would help identify those obese individuals who are at high risk of CCAP.

Epidemiological studies have identified obesity as a risk factor for an expanding set of chronic diseases, including CVD, DM, dyslipidaemia, hypertension and chronic kidney disease.[30 31] Recent studies have reported that obesity increased the risk of subclinical carotid AS.[32–34] In this study, 28.64% obese participants had CCAP. The best predictive marker of CCAP was high LDL-C:HDL-C ratio, and the serum LDL-C and HDL-C also discriminated CCAP. The demonstrated high prevalence of CCAP in obese individuals expands the findings made in the previous population-based studies in which the association of obesity with atherosclerotic progression has been proven.[33 35 36]

The causal relationship between LDL-C and increased risk of CAD has been well established. However, even aggressive reduction in LDL-C using statin agents still leaves considerable residual risk of CAD events. Early epidemiological studies demonstrated lower HDL-C level is related with an increased risk of CAD. However, some recent studies failed to show significant association of HDL-C levels and outcomes for patients with CAD on intensive lipid therapy or patients with CAD undergoing elective coronary artery bypass grafting. Nowadays, the focus of the HDL-C hypothesis has been directed toward the function of HDL-C.

The LDL-C:HDL-C ratio has recently been recognised as a new biometric that has clinical relevance in AS diseases. In the Helsinki Heart Study analyses, LDL-C:HDL-C ratio was shown to be the best single predictor of cardiac events.[37] Kunutsor et al[25] conducted a study investigating the association between LDL-C:HDL-C ratio and sudden cardiac death (SCD). High LDL-C:HDL-C ratio was found to be independently associated with an increased risk of SCD. There was approximately a twofold increase in the risk of SCD comparing the top versus bottom quintile of serum

**Table 3** Clinical characteristics of study population stratified by LDL-C:HDL-C ratio quartiles

| Characteristics | LDL-C:HDL-C ratio quartiles | | | | P values |
| --- | --- | --- | --- | --- | --- |
| | First (<2.07) n=362 | Second (≥2.07–<2.52) n=362 | Third (≥2.52–<3.25) n=362 | Fourth (≥3.25) n=363 | |
| Gender | | | | | 0.160 |
| Women, n (%) | 189 (52.21) | 161 (44.48) | 173 (47.79) | 165 (45.45) | |
| Men, n (%) | 173 (47.79) | 201 (55.52) | 189 (52.21) | 198 (54.55) | |
| Age, years | 52.00 (43.00, 64.00) | 54.00 (45.00, 62.00) | 52.00 (45.00, 63.00) | 53.00 (46.00, 63.00) | 0.370 |
| Body mass index, kg/m$^2$ | 30.21 (29.04, 32.38) | 29.74 (28.67, 31.25) | 30.11 (28.99, 32.45)* | 30.49 (29.05, 32.69)† | <0.001 |
| Smoking, n (%) | 42 (11.60) | 62 (17.13) | 58 (16.02) | 84 (23.14)‡ | 0.001 |
| Hypertension, n (%) | 118 (32.60) | 102 (28.18) | 109 (30.11) | 115 (31.68) | 0.590 |
| Antihypertensive drugs, n (%) | 83 (22.93) | 69 (19.06) | 80 (22.10) | 79 (21.76) | 0.613 |
| Diabetes mellitus, n (%) | 53 (14.64) | 60 (16.57) | 88 (24.31)*§ | 123 (33.88)†‡¶ | <0.001 |
| Antidiabetic agents, n (%) | 25 (6.91) | 35 (9.67) | 41 (11.33) | 46 (12.67) | 0.061 |
| Dyslipidaemia, n (%) | 79 (21.82) | 113 (31.22)** | 180 (49.72)*§ | 326 (89.81)†‡¶ | <0.001 |
| Statins, n (%) | 31 (8.56) | 42 (11.60) | 40 (11.05) | 33 (9.09) | 0.455 |
| CCAP, n (%) | 35 (9.67) | 72 (19.89)** | 108 (29.83)*§ | 200 (55.10)†‡¶ | <0.001 |
| Maximal CCA-IMT, mm | 1.00 (0.80, 1.20) | 1.00 (0.80, 1.30) | 1.00 (0.80, 1.40) | 1.30 (0.90, 1.60)†‡¶ | <0.001 |
| Mean CCA-IMT, mm | 0.80 (0.60, 1.00) | 0.80 (0.60, 1.00) | 0.80 (0.60, 1.10) | 1.00 (0.70, 1.30)†‡¶ | <0.001 |

Date were compared by Mann-Whitney tests or $\chi^2$ tests.
*P<0.0083 when comparing the second quartile versus the third quartile.
†P<0.0083 when comparing the second quartile versus the fourth quartile.
‡P<0.0083 when comparing the first quartile versus the fourth quartile.
§P<0.0083 when comparing the first quartile versus the third quartile.
¶P< 0.0083 when comparing the third quartile versus the fourth quartile.
**P<0.0083 when comparing the first quartile versus the second quartile.
CCA-IMT, common carotid artery intima-media thickness; CCAP, common carotid atherosclerotic plaque; HDL-C, high-density lipoprotein cholesterol; LDL-C, low-density lipoprotein cholesterol.

LDL-C:HDL-C ratio. Another study reported elevated LDL-C:HDL-C ratio was related with the progression of coronary artery lesions.[25 38]

Recently, several investigators have examined the utility of LDL-C:HDL-C ratio in predicting carotid AS. However, contrasting results have been demonstrated. Enomoto et al[22] conducted a periodic epidemiological survey to investigate the relationship between lipid profiles and the progression of common carotid IMT. They found LDL-C:HDL-C ratio was a better marker in predicting IMT progression than HDL-C or LDL-C alone during an 8-year follow-up survey. Katakami et al[23] explored the relationships between various lipid parameters including lipid ratios and carotid AS in 934 patients with T2DM having no obvious atherosclerotic diseases in Japan. They concluded that LDL-C:HDL-C ratio was positively associated with carotid atherosclerotic plaques. Wu et al[39] surveyed the impact of LDL-C:HDL-C ratio on CIMT among 1579 residents aged 40–74 years in northern Taiwan. The results demonstrated LDL-C:HDL-C ratio was strongly correlated with CIMT in both men and women, indicating that LDL-C:HDL-C ratio was an important determinant of increased CIMT.

However, Nimkuntod et al[40] failed to identify any associations between LDL-C:HDL-C ratio and early carotid AS CIMT carotid plaques as well as carotid plaque type. The small sample of the study may possibly explain the discrepancy. In our study, we found obese individuals with CCAP had significantly higher prevalence of dyslipidaemia. Several pathophysiological processes have been proposed to explain the mechanisms of dyslipidaemia in obese population. First, dyslipidaemia was mainly driven by the increased lipid production and hydrolysis of TG. Second, higher plasma level of free fatty acid may also contribute to dyslipidaemia. According to our findings, high LDL-C:HDL-C ratio is the strongest predictor of CCAP after adjusting for major risk factors, thus indicating that the high LDL-C:HDL-C ratio may influence the progression of CCAP through pathways beyond traditional risk factors.

Our finding is consistent with previous studies, and revealed a positive association between high LDL-C:HDL-C ratio and CCAP in Uygur obese adults. While the exact mechanism of the superiority of LDL-C:HDL-C ratio to either individual LDL-C or HDL-C remains to be elucidated, several explanations could be suggested. First, LDL-C:HDL-C ratio represents the proportion between the atherogenic and

**Table 4** Logistic regression analysis of the risk factors for CCAP

| Variables | Univariate | | | Multivariate | | |
|---|---|---|---|---|---|---|
| | OR | 95% CI | P value | OR | 95% CI | P values |
| Gender | | | | – | | |
| Female | Reference | | | – | | |
| Male | 1.085 | 0.864 to 1.364 | 0.482 | – | | |
| Age | | | 0.013 | – | | 0.040 |
| <45 years | Reference | | | Reference | | |
| 45–54 years | 1.631 | 1.178 to 2.257 | 0.003 | 1.425 | 0.999 to 2.034 | 0.051 |
| 55–64 years | 1.485 | 1.048 to 2.104 | 0.026 | 1.584 | 1.081 to 2.322 | 0.018 |
| 65–74 years | 1.625 | 1.132 to 2.333 | 0.009 | 1.642 | 1.101 to 2.451 | 0.015 |
| ≥75 years | 2.288 | 1.220 to 4.292 | 0.010 | 2.375 | 1.176 to 4.799 | 0.016 |
| Body mass index | 1.068 | 1.024 to 1.114 | 0.002 | 1.058 | 1.010 to 1.108 | 0.018 |
| Smoking | 1.559 | 1.167 to 2.081 | 0.003 | 1.407 | 1.019 to 1.943 | 0.038 |
| Hypertension | 0.936 | 0.730 to 1.200 | 0.600 | – | | |
| Diabetes mellitus | 2.082 | 1.606 to 2.697 | <0.001 | 1.590 | 1.196 to 2.113 | 0.001 |
| TG | 0.935 | 0.817 to 1.070 | 0.327 | – | | |
| TC | 1.512 | 1.323 to 1.728 | <0.001 | 1.437 | 1.244 to 1.659 | <0.001 |
| LDL-C | 3.012 | 2.274 to 3.917 | <0.001 | – | | |
| HDL-C | 0.222 | 0.145 to 0.340 | <0.001 | – | | |
| LDL-C:HDL-C ratio | | | <0.001 | | | <0.001 |
| First quartile (<2.07) | Reference | | | Reference | | |
| Second quartile (≥2.07–<2.52) | 2.320 | 1.503 to 3.579 | <0.001 | 2.184 | 1.404 to 3.399 | 0.001 |
| Third quartile (≥2.52–<3.25) | 3.973 | 2.623 to 6.016 | <0.001 | 3.556 | 2.333 to 5.421 | <0.001 |
| Fourth quartile (≥3.25) | 11.464 | 7.644 to 17.193 | <0.001 | 9.355 | 6.181 to 14.157 | <0.001 |
| High sensitivity C-reactive protein | 1.006 | 0.966 to 1.049 | 0.760 | – | | |
| Uric acid | 1.053 | 0.971 to 1.141 | 0.214 | – | | |
| Antihypertensive drugs | 0.884 | 0.667 to 1.172 | 0.390 | – | | |
| Antidiabetic agents | 1.369 | 0.955 to 1.963 | 0.088 | – | | |
| Statins | 0.831 | 0.561 to 1.229 | 0.353 | – | | |

CCAP, common carotid atherosclerotic plaque; HDL-C, high-density lipoprotein cholesterol; LDL-C, low-density lipoprotein cholesterol; TC, total cholesterol; TG, triglyceride.

antiatherosclerotic lipoproteins, which has greater predictive value for atherosclerotic changes. Second, LDL-C:HDL-C ratio was closely correlated with the distribution of HDL subclasses. With an increase in the LDL-C:HDL-C ratio, there was a general shift toward smaller-sized HDL particles, implying that the maturation process of HDL was blocked, which may contribute to the progression of AS.[41] Third, high LDL-C:HDL-C ratio may lead to the fluctuations in the composition of atherosclerotic plaques. High LDL-C:HDL-C ratio may be associated with the development of large lipid-rich necrotic core.

Our previous work described the cut-off value of LDL-C:HDL-C ratio for assessing CVD risk factors was 2.5 among Uygur adults.[42] In the present study, participants with LDL-C:HDL-C ratio ≥2.07 had a significantly higher risk of CCAP, which was higher than that obtained by Kurabayashi *et al*.[43] They found the cut-off value of LDL-C:HDL-C ratio for initiating secondary prevention

of AS was >1.5, and the cut-off ratio for initiating primary prevention of AS was >2.0. The discordance between our results may be due to different characteristics between our population and theirs. The participants recruited into our study were obese individuals, and the mean level of LDL-C is significantly higher than those previous studies, which may help to explain the high LDL-C:HDL-C ratio in our study. Second, the distinct ethnic origin, diet, lifestyles, living habits and living environments that are specific to Uygur population may contribute to these discrepancies. Further studies are needed to confirm our finding of high LDL-C:HDL-C ratio.

Several limitations of this study should be considered. First, our findings are based on a population from a single centre, and should be verified in other larger populations. Second, this is a cross-sectional analysis, which limits the temporal interpretation of predictive measurements. Third, we only measured baseline LDL-C:HDL-C ratio,

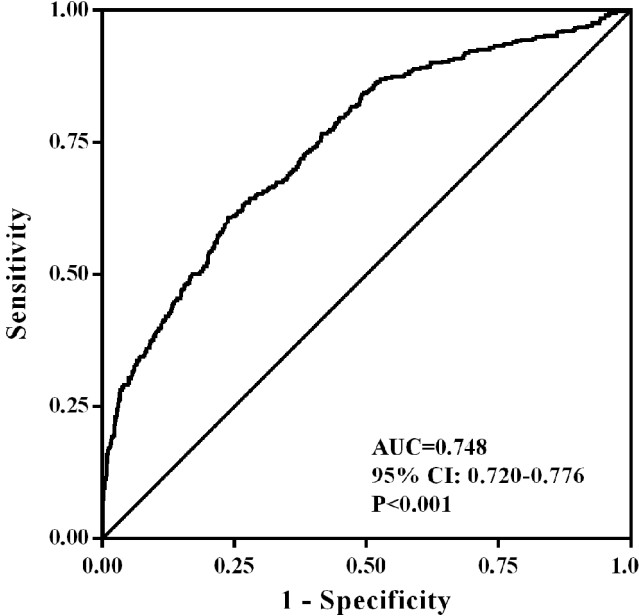

**Figure 2** Receiver operating characteristic curve analysis of low-density lipoprotein cholesterol:high-density lipoprotein cholesterol (LDL-C:HDL-C) ratio for predicting common carotid atherosclerotic plaque. The area under the curve (AUC) of LDL-C:HDL-C ratio was 0.748 (95 % CI 0.720 to 0.776, p<0.001).

and the relations between LDL-C:HDL-C ratio changes and outcomes could not be ascertained. In future studies, LDL-C:HDL-C ratio during the follow-up period should also be examined. Fourth, the absence of a health control group should be considered as the main constraint of our study. Whether the results can be applied to general obese population remains to be investigated. Finally, statins are known to affect changes in the LDL-C:HDL-C ratio.[44] About 10% participants had taken statin medication in our study, so we cannot evaluate its influence on lipid ratio changes and the progression of CCAP.

In summary, our study demonstrated that high LDL-C:HDL-C ratio is independently associated with increased risk of CCAP among Uygur obese adults in Xinjiang, China. LDL-C:HDL-C ratio may help identify obese individuals who are at high risk of CCAP and who may benefit from intensive LDL-lowering therapy. Further studies are needed to investigate the predictive value of LDL-C:HDL-C ratio in the development of carotid plaque components.

**Acknowledgements** We thank all of the participants for their cooperation in this study and the staff of the Xinjiang Key Laboratory of Cardiovascular Disease Research for their dedicated assistance in sample collection.

**Contributors** Y-NY, QZ and FL conceived and designed the study. QZ, Y-HW, H-ML and J-YL performed the experiments and collected the data. Y-NY, Y-TM and X-ML completed the quality of data. QZ and FL performed the statistical analysis and wrote the paper. All authors read and approved the final manuscript.

**Funding** This work was supported by the National Natural Science Foundation of China (No. 81460069, 81700315, U1503322), Science and Technology Agency of Xinjiang Uygur Autonomous Region (No. 2016E02072) and Postgraduate Scientific Research Project of Xinjiang Uygur Autonomous Region (No. XJGRI2017081).

**Competing interests** None declared.

**Patient consent** Obtained.

**Ethics approval** The study was carried out in accordance with the declaration of Helsinki and the study protocol was approved by the Ethical Committee of First Affiliated Hospital of Xinjiang Medical University.

**Provenance and peer review** Not commissioned; externally peer reviewed.

**Data sharing statement** No additional data are available.

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
