## [Reviewer comments · BMJ Open]

ARTICLE DETAILS

TITLE (PROVISIONAL)	LDL-C/HDL-C ratio and common carotid plaque in Xinjiang Uygur obese adults: a cross-sectional study
AUTHORS	Zhao, Qiang; Liu, fen; Wang, Ying-Hong; Lai, Hong-Mei; Zhao, Qian; Luo, Jun-Yi; Ma, Yi-Tong; Li, Xiao-Mei; Yang, Yi-Ning

VERSION 1 – REVIEW

REVIEWER	Hiroaki Suzuki Department of Internal Medicine, Faculty of Medicine, University of Tsukuba, Japan
REVIEW RETURNED	28-Mar-2018

GENERAL COMMENTS	The author reported that the associations between a LDL-C/HDL-C ratio and common carotid atherosclerosis in an obese Chinese Uygur population. They found that LDL-C/HDL-C ratio was an independent predictor for carotid atherosclerosis after adjustments for traditional cardiovascular risk factors including plasma lipids levels and that a LDL-C/HDL-C ratio of 2.86 was a cut point to determine common carotid atherosclerotic plaque (CCAP) in their population. Although they tested sufficient numbers of subjects to determine a cut point, it seems that there are few noteworthy new findings in this paper. There are many reports on the association between LDL-C/HDL-C ratio and CCAP. Major point 1. The authors should describe the method to measure LDL-C. Did the authors measure LDL-C using a direct assay or using Friedewald formula?2. Plasma TG levels usually have non-normal distribution. However, TG levels are expressed as mean \pm SD. Did the author checked the distribution?3. Polak JF and colleagues reported that the maximum IMT of the internal carotid artery was superior to that of the common carotid artery to predict future cardiovascular events (Polak JF, et al. NEJM 2011;365:213-21). If the authors measured the maximum IMT of the internal carotid artery, it is better to analyze the data on the association between LDL-C/HDL-C ratio and internal carotid artery plaque. Minor point 1. Is “antihypertensive” a typographical error for “anti-diabetic”? (P6 line 4)
--

REVIEWER	Jari A. Laukkanen University of Jyväskylä, Jyväskylä, Finland
REVIEW RETURNED	26-May-2018

GENERAL COMMENTS	The study was designed to explore the relationship between the level of LDL-C/HDL-C ratio and common carotid atherosclerotic plaque (CCAP) among Uygur obese adults in Xinjiang. The authors reported a cross-sectional study undertaken at the First Affiliated Hospital of Xinjiang Medical University from 1 January 2014 to 31 December 2016.  1. Please check if LDL-C/HDL-C ratio is possible to include into the model together with each of them separately. 2. If CRP was measured, you could show if inclusion of CRP would have effect on the results. Low grade inflammation may contribute the development of atherosclerosis. 3. Tables, you should use accuracy of the numbers which is relevant (decimal places) 4. What was the rationale to use the cut-off of 2.86 for LDL-C/HDL-C ratio? It should have been carefully justified? Please consider using i.e. quintiles or quartiles of LDL-C/HDL-C ratio 5. In the discussion, there are some long paragraphs (on page 13), you could try to modify this part of the discussion. 6. There would be previous data on HDL and LDL ratio and the risk of sudden cardiac death which might be helpful to revise this manuscript further (J Atheroscler Thromb. 2017 Jun 1;24(6):600-608). Please consider to add.
--

VERSION 1 – AUTHOR RESPONSE

Response to Reviewer 1

(1). The authors should describe the method to measure LDL-C. Did the authors measure LDL-C using a direct assay or using Friedewald formula?

Response: I have described the method to measure LDL-C in the Lipid profile analyses section. It has been presented as follows: "Triglycerides (TG), total cholesterol (TC), LDL-C and HDL-C levels were directly measured using the homogeneous enzymatic colorimetric assay (Roche Diagnostics) on an automatic analyzer (Cobas 8000)." (Page 6, Lines 21-23).

(2). Plasma TG levels usually have non-normal distribution. However, TG levels are expressed as mean \pm SD. Did the author checked the distribution?

Response: Thanks for your suggestion, I checked the distribution of TG levels again and found the distribution of TG levels is non-normal. I have amended the description of TG levels in the Table 2 (Laboratory and carotid ultrasonography parameters of study population according to CCAP status), which has been described as follows: "TG, mmol/L: overall 1.60 (1.11, 2.15); CCAP group 1.53 (1.14, 2.00); non-CCAP group 1.62 (1.10, 2.16); $P=0.375$ ". (Page 10, Table 2, Line 3).

(3) Polak JF and colleagues reported that the maximum IMT of the internal carotid artery was superior to that of the common carotid artery to predict future cardiovascular events (Polak JF, et al. NEJM 2011;365:213-21). If the authors measured the maximum IMT of the internal carotid artery, it is better to analyze the data on the association between LDL-C/HDL-C ratio and internal carotid artery plaque.

Response : Thanks for your suggestion. We noticed that there were some studies examining the association between IMT of the internal carotid artery (ICA-IMT) and CAD. In fact, we have measured ICA-IMT in a part of participants. However, obtaining adequate images of the ICA segments was difficult in some very obese participants, and we failed to measure the maximal ICA-IMT. Moreover, we could only measure the IMT at the very beginning of ICA in some participants, and we cannot identify the ICA-IMT. The data of maximum IMT of the ICA was incomplete, so we did not analyze the association between LDL-C/HDL-C ratio and internal carotid artery plaque.

(4) Minor point

1. Is “antihypertensive” a typographical error for “anti-diabetic”? (P6 line 4)

Response: Thanks for your suggestion. I noticed it either. It is a typographical error which has been changed into “antidiabetic” (Page 6, Line 12).

Furthermore, all the numbers in the tables have been corrected to two decimal places (as shown in Table 1, Table 2, Table 3 and Table 4).

Response to Reviewer 2

(1). Please check if LDL-C/HDL-C ratio is possible to include into the model together with each of them separately.

Response: Thanks for your suggestion. When we reviewed the method of statistical analyses, we found it is improper to include the LDL-C/HDL-C ratio into the multivariate model together with each of them separately. Because LDL-C/HDL-C ratio integrates both LDL-C and HDL-C, and is strongly correlated with the LDL-C and HDL-C, only LDL-C/HDL-C ratio was involved into multivariate analysis in the present study. In the present study, LDL-C/HDL-C ratio was included into the CCAP univariate/multivariate model as a quartile categorical variable. The paragraph of **Association of LDL-C/HDL-C ratio and CCAP** has been presented as follows: “To determine the linear trends of the risk, we explored the ORs of CCAP by LDL-C/HDL-C ratio quartile groups, with the first quartile serving as the reference category. Entering LDL-C/HDL-C ratio as quartiles to the CCAP univariable model revealed that the top quartile (fourth quartile, ≥ 3.25) had an OR of 11.464 (95% CI: 7.644 - 17.193) compared with the bottom quartile (first quartile, < 2.07) ($P < 0.001$). LDL-C (OR: 3.012, 95 %

CI: 2.274 - 3.917, $P < 0.001$) and HDL-C (OR: 0.222, 95 % CI: 0.415 - 0.340, $P < 0.001$) were also found to be independent predictors of CCAP in univariate analysis. Other significant correlates of CCAP were age (> 45 years, every 10 years), BMI, smoking, DM, and TC (all $P < 0.05$). Because LDL-C/HDL-C ratio integrates both LDL-C and HDL-C, and is strongly correlated with the LDL-C and HDL-C, only LDL-C/HDL-C ratio was involved into multivariate analysis.

After adjusting for age, BMI, smoking, presence of DM, and mean TC level, the association between LDL-C/HDL-C ratio and CCAP still remained significant (OR [95% CI]: second quartile, 2.184 [1.404 - 3.399]; third quartile, 3.556 [2.333 - 5.421]; top quartile, 9.355 [6.181 - 14.157]). The multivariable-adjusted ORs of CCAP increased continuously and linearly, and statistical significance was observed from the second quartile of LDL-C/HDL-C ratio. The top LDL-C/HDL-C ratio quartile had an OR of 9.355 (95% CI: 6.181 - 14.157) compared with the bottom quartile ($P < 0.001$) after adjustment for age, BMI, smoking, DM and TC.” (Page 12, Lines 9-27).

(2). If CRP was measured, you could show if inclusion of CRP would have effect on the results. Low grade inflammation may contribute the development of atherosclerosis.

Response: We measured the serum high-sensitivity C-reactive protein (hs-CRP) levels, and the results of hs-CRP have been put in the table 2. However, there was no significant difference in serum hs-CRP levels between the two groups, describing as “(2.71 [1.33, 4.35] mg/mL vs. 2.64 [1.25, 4.44] mg/mL; $P = 0.734$).” (Page 10, Table 2, Line 9).

A moderately increased serum level of hs-CRP is well established as a marker of cardiovascular disease risk. However, some recent studies relating hs-CRP level to the progression of atherosclerosis have showed conflicting results. Zacho J (Pubmed:18971492) and Elliott P (PMID:19567438) have not supported the causal association of hs-CRP with atherosclerosis, respectively. In the present study, we failed to demonstrate that serum hs-CRP correlates with CCAP. It may be due to the small sample, different coexisting risk factors and that the participants in the study are not representative of subjects in population-based epidemiological studies.

(3) Tables, you should use accuracy of the numbers which is relevant (decimal places)

Response : Thanks for your suggestions, all the numbers in the tables have been corrected to two decimal places (as shown in Table1, Table2, Table3 and Table 4).

(4) What was the rationale to use the cut-off of 2.86 for LDL-C/HDL-C ratio? It should have been carefully justified? Please consider using i.e. quintiles or quartiles of LDL-C/HDL-C ratio.

Response: We rechecked paper again and found it is improper to use the cut-off value of LDL-C/HDL-C ratio to predict the risk of CCAP. Therefore, we deleted the part about cut-off value of LDL-C/HDL-C ratio in the last paragraph of the result. We only described the AUC values for LDL-C/HDL-C ratio. Furthermore, subjects were classified according to LDL-C/HDL-C ratio quartiles according to your suggestion. LDL-C/HDL-C ratio was included into the CCAP univariate/multivariate logistic regression model as a quartile categorical variable, as shown from the fifth to the eighth paragraph of the result and Table 3 as well as Table 4. (From Page 11, Line 6 to Page 14, Line 1).

(5) In the discussion, there are some long paragraphs (on page 13), you could try to modify this part of the discussion.

Response: Thanks for your advice, we have modified the long paragraphs on page 14-16. (As shown from the third to the seventh paragraph of the discussion, from Page 14, Line 23 to Page 16, Line 17).

(6) There would be previous data on HDL and LDL ratio and the risk of sudden cardiac death which might be helpful to revise this manuscript further (J Atheroscler Thromb. 2017 Jun 1;24(6):600-608). Please consider to add.

Response: Thanks for your advice. The paper on the association of LDL-C/HDL-C ratio with sudden cardiac death was valuable and helpful for us to revise our manuscript. We have revised the manuscript after reread that paper and added it as an important reference of this manuscript to support our thesis. (As shown in the fourth paragraph of the discussion. Page 15, Lines 4-10).

VERSION 2 – REVIEW

REVIEWER	Hiroaki Suzuki Department of Internal Medicine (Endocrinology and Metabolism), Faculty of Medicine, University of Tsukuba, Japan
REVIEW RETURNED	29-Jul-2018

GENERAL COMMENTS	I have no comments on the revised article.
--

REVIEWER	Jari A. Laukkanen University of Jyväskylä, Jyväskylä, Finland
REVIEW RETURNED	03-Jul-2018

GENERAL COMMENTS	The authors have revised the manuscript as requested. No additional comments.
---